# Ibero-American Consensus Review and Incorporation of New Biomarkers for Clinical Practice in Colorectal Cancer

**DOI:** 10.3390/cancers15174373

**Published:** 2023-09-01

**Authors:** Carlos Eduardo Bonilla, Paola Montenegro, Juan Manuel O’Connor, Ovidio Hernando-Requejo, Enrique Aranda, José Pinto Llerena, Alejandra Llontop, Jorge Gallardo Escobar, María del Consuelo Díaz Romero, Yicel Bautista Hernández, Begoña Graña Suárez, Emilio J. Batagelj, Ahmad Wali Mushtaq, Jesús García-Foncillas

**Affiliations:** 1Fundación CTIC—Centro de Tratamiento e Investigación sobre Cáncer, Bogotá 1681442, Colombia; 2Institución AUNA OncoSalud e Instituto Nacional de Enfermedades Neoplásicas, Lima 15023, Peru; 3Instituto Alexander Fleming, Buenos Aires C1426, Argentina; juanmanuel.oconnor@gmail.com; 4Centro Integral Oncológico Clara Campal, HM Hospitales, 28660 Madrid, Spain; ohernando@hmhospitales.com; 5Departamento de Oncología Médica, Hospital Reina Sofía, IMIBIC, UCO, CIBERONC, 14004 Cordoba, Spain; earandaa@seom.org; 6Instituto Oncológico Nacional, Panama City 0816, Panama; 7Instituto de Oncología Ángel H. Roffo, Ciudad Autónoma de Buenos Aires C1437FBG, Argentina; 8Clínica Las Condes, Santiago de Chile 7591047, Chile; 9Instituto Nacional de Cancerología, Mexico City 14080, Mexico; 10Hospital General de México, Mexico City 06720, Mexico; yiselbautista@prodigy.net.mx; 11Servicio de Oncología Médica, Hospital Universitario de A Coruña, Servicio Galego de Saúde (SERGAS), 15006 A Coruña, Spain; begona.grana.suarez@sergas.es; 12Hospital Militar Central de Buenos Aires, Buenos Aires C1426, Argentina; 13Servicio de Oncología, Hospital Eugenio Espejo, Quito 170136, Ecuador; 14Hospital Universitario Fundación Jiménez Díaz, Universidad Autónoma de Madrid, 28040 Madrid, Spain

**Keywords:** biomarkers, colorectal, genomic, screening, multidisciplinary

## Abstract

**Simple Summary:**

The management of colorectal cancer has improved significantly due to advances in molecular techniques. Thanks to these techniques, it has been possible to identify markers in these tumors, which allows for the use of specific treatments. However, research on this subject is evolving so rapidly that previous information is quickly rendered obsolete. In addition, these techniques offer a large amount of information, which makes them difficult to interpret. The aim of this article is to describe how these markers are determined from a practical point of view.

**Abstract:**

Advances in genomic technologies have significantly improved the management of colorectal cancer (CRC). Several biomarkers have been identified in CRC that enable personalization in the use of biologic agents that have shown to enhance the clinical outcomes of patients. However, technologies used for their determination generate massive amounts of information that can be difficult for the clinician to interpret and use adequately. Through several discussion meetings, a group of oncology experts from Spain and several Latin American countries reviewed the latest literature to provide practical recommendations on the determination of biomarkers in CRC based on their clinical experience. The article also describes the importance of looking for additional prognostic biomarkers and the use of histopathology to establish an adequate molecular classification. Present and future of immunotherapy biomarkers in CRC patients are also discussed, together with several techniques for marker determination, including liquid biopsy, next-generation sequencing (NGS), polymerase chain reaction (PCR), and fecal immunohistochemical tests. Finally, the role of Molecular Tumor Boards in the diagnosis and treatment of CRC is described. All of this information will allow us to highlight the importance of biomarker determination in CRC.

## 1. Introduction

Colorectal cancer (CRC) is a heterogeneous disease that results from the interaction of multiple genetic modifications and exogenous factors such as diet, lifestyle, and microbiome [1]. Despite this heterogeneity, advances in genomic technologies have significantly improved the management of cancer patients. Identification of a driver gene mutation or other biomarkers can lead to specific targeted therapies, resulting in precision and personalized medicine that can improve the clinical outcomes of these patients [2]. The implementation of precision medicine and molecular genetic testing for cancer patients remains an ongoing educational process for physicians in both hospitals and educational centers [3,4]. The identification of potential new biomarker-based pharmacological treatments and therapeutic studies depends largely on the experience and knowledge of the medical team involved in the treatment of these patients.

Although numerous guidelines have been published on biomarker determination in CRC [1,5,6], the large amounts of rapidly evolving literature on precision oncology means that many of them are often outdated, challenging clinicians to be up to date in the biomarker field and critically examine all of this information to provide patients with the best possible molecular counseling [7,8]. In addition, genomic technologies generate massive amounts of information that can be difficult for the clinician to interpret and use adequately, resulting in a large disparity between clinical knowledge and genetic potential in cancer care [3].

Through several discussion meetings, a group of oncology experts from Spain and several Latin American countries reviewed the latest literature on the topic to provide practical recommendations on the determination of biomarkers in CRC based on their clinical experience.

## 2. Markers in CRC

There are several types of markers in CRC that have value in diagnosis, prognosis, and prediction of therapeutic response, disease monitoring, and recurrence. These markers include genetic or molecular alterations and even the anatomic location of the primary tumor.

CRC has been considered a successful model for the determination of genetic biomarkers in oncology such as *KRAS*, *NRAS*, *BRAF* mutations, *HER2* amplification, and microsatellite instability high (MSI-H) or mismatch repair deficiency (dMMR) [1,9]. Genomic alterations influence treatment selection, and since targeted therapy for the treatment of advanced or metastatic CRC (mCRC) is essential, some international guidelines recommend assessing the mutational status of the genes involved in tumor development [1,10,11,12]. Table 1 shows brief and practical recommendations for the determination of biomarkers involved in CRC. Appendix A shows the prognostic and predictive value of each biomarker [1,2,9,10,11,12,13,14,15,16,17,18,19,20,21,22,23,24,25,26,27,28,29,30,31,32,33,34,35,36,37,38,39,40,41,42,43,44,45,46,47,48,49,50,51,52,53,54,55,56,57,58,59,60,61,62].

Along with these biomarkers, it is also important to highlight the role of homologous recombination deficiency (HRD) in CRC. HRD is caused by loss of the *BRCA*1/2 allele during carcinogenesis or by other genomic aberrations, which increases vulnerability to therapies such as platinum that crosslink DNA and cause DNA double-strand breaks, exceeding the ability to repair DNA damage. HRD-related somatic mutations are found in 13.8% of patients with CRC and are enriched in MSI-H, right-sided, and *BRAF* mutant cancers [63].

Although the anatomic location of a primary tumor across the colon is not a biomarker per se, it has been correlated with the prognosis of patients with CRC. Left-sided tumors may exhibit superior survival compared with those that are right-sided [64]. Moreover, it has been reported that right-sided tumors exhibit more CDX2-negative tumors compared with left-sided tumors. Lacking this transcription factor has been associated with several adverse prognostic variables and a high pathological grade. CDX2-negative CRCs are associated with a higher risk of recurrence and seem to benefit from adjuvant chemotherapy compared with CDX2-positive colorectal tumors [64].

Despite the fact that many of the biomarkers described are not routinely determined, it is important to mention them by emphasizing their predictive and prognostic value, even if only briefly, as they add value to the classical approach of biomarkers so far considered, which have been reduced classically only to RAS, BRAF, and MSI.

## 3. Histopathology as a Disease Marker in CRC

Histopathology has an important role in establishing an adequate molecular classification of CRC. Thanks to this classification, patients can be stratified according to their risk of metastasis or recurrence, which helps to select the most appropriate treatment in each case. Several classification systems and histopathological procedures are described below.

### 3.1. Consensus Molecular Subtypes (CMS)

Although the tumor, node, and metastasis (TNM) staging system remains the gold standard for the stratification of patients with CRC, the heterogeneity of CRC points to the need for additional prognostic biomarkers [12,65]. The CMS classification of CRC is based on a combined analysis of RNA sequencing data from multiple international cohorts and includes molecular factors, tumor stroma, and signaling pathways for personalized systemic therapy [66,67]. However, this classification does not yet have a translation to clinical practice.

### 3.2. Tumor Budding (TB)

TB is an adverse histologic feature associated with poor prognosis that arises because of the loss of adhesion of the neoplastic cells of the tumor, which leads to the initiation of the metastatic process in units of 1 to ≤4 cells called buds [10,11]. According to the International Tumor Budding Consensus Conference (ITBCC), there may be BD1 (0–4 buds, low budding), BD2 (5–9 buds, intermediate budding), or BD3 (≥10 buds, high budding) [65]. Both BD2 and BD3 are risk factors for nodal metastases in patients with pT1 (stage I) CRC, whereas only BD3 is associated with an increased risk of recurrence and death in those with stage II CRC. In particular, BD3 tumors could be candidates for adjuvant therapy [68]. TB has been included in major staging systems/guidelines as an additional prognostic factor [68,69]. TB could potentially be considered relevant in the following scenarios: (1) to determine the risk of node metastasis in patients with early-stage CRC and thus report the need for radical surgery; (2) to identify patients with high-risk stage II colon cancer, a potential indication for adjuvant therapy; and (3) as an indicator of metastasis and lack of response to neoadjuvant therapy if detected in pretreatment biopsies [15]. According to this expert consensus, TB reporting is recommended but is not a required element.

### 3.3. ImmunoScore^®^

ImmunoScore is a scoring system reported as percentiles of CD3+ and CD8+ immune cell densities in predefined regions of the tumor samples using software, with the aim of assessing the prognostic value of patients with stage III colon cancer, as well as its predictive value for response to adjuvant chemotherapy in these patients [70]. Patients with a high ImmunoScore will benefit more from chemotherapy in terms of risk of recurrence. Those with a high ImmunoScore have a low risk of recurrence and prolonged time to recurrence, overall survival, and disease-free survival [10,11,70]. According to experts’ consensus, there are no studies that suggest applying this scoring system routinely in clinical practice, although it is an auxiliary tool for predicting response to adjuvant therapy. It is available in some cases but is not approved.

### 3.4. The Circumferential Resection Margin (CRM)

The CRM should be recorded in all non-peritonealized resections, as poor prognosis is associated with compromised CRM [71]. A positive margin is defined as (1) a tumor < 1 mm from the sectioned margin, (2) a tumor < 2 mm from the sectioned margin, and (3) cells that are in contact with the electrocautery-affected area [10,11].

### 3.5. The Mucinous Component

The mucinous component is defined as a tumor in which more than 50% of the lesion consists of extracellular pools of mucin, besides being phenotypically distinct from adenocarcinoma not otherwise specified [72]. Mucinous differentiation accounts for 5–15% of colorectal adenocarcinomas. This subtype of CRC responds poorly to chemoradiotherapy and has a poor prognosis. The genetic origins of mucinous colorectal adenocarcinoma are predominantly associated with BRAF, MSI, and CIMP pathways [73]. According to experts’ consensus, in pathology reports it is important to indicate presence of this component, as well as the corresponding percentage. Mucinous adenocarcinoma (MAC) with a signet ring cell component > 50% should be classified as signet ring cell carcinoma [74]. This is a rare histologic subtype of adenocarcinomas with a poor prognosis, usually due to the molecular alterations described in mucinous CRC and diagnosis in advanced stages [75,76]. The experts suggest reporting the presence of the signet ring cell component separately from the mucinous component.

## 4. Immunotherapy Biomarkers in CRC

Checkpoint inhibitors have changed the treatment paradigm in several types of solid tumors. The selection of patients potentially susceptible to responding to these treatments requires robust predictive biomarkers. Some of the most important biomarkers related to immunotherapy in CRC are described below.

### 4.1. Human Leukocyte Antigen Class I (HLA-I)

HLA-I plays a critical role in antigen presentation to T lymphocytes, including tumor antigens. These molecules are frequently lost in CRC, resulting in immune escape to cytotoxic T lymphocytes during the natural history of cancer development [77]. This phenomenon has important implications when T-cell-mediated immunotherapy is applied in cancer patients [77]. The absence of HLA-I expression allows tumor cells to avoid recognition by cytotoxic T lymphocytes, whereas natural killer (NK) cells are activated [78].

### 4.2. Tumor Microenvironment (TME)

Tumors comprise epithelial cells and stroma, both of which configure the TME made up of distinct and interacting cell populations [79,80]. Abnormalities of the extracellular matrix relieve the behavioral regulation of stromal cells and promote angiogenesis and tumor inflammation, resulting in resistance to immunotherapy in the TME [81]. Cancer-associated fibroblasts (CAFs) or myofibroblasts are responsible for the production and regulation of stroma in tumor tissue, depositing the same matrix components that make up tumor connective tissue, a process that is termed desmoplastic reaction (DR). CAFs modulate cancer cells through the production of growth factors [79]. Histological classification of DR provides important prognostic information that could contribute to the selection of patients with stage II colon cancer who would benefit from postoperative adjuvant therapy [82].

### 4.3. Transforming Growth Factor Beta (TGF-β)

TGF-β is considered a tumor suppressor cytokine. However, TGF-β may transform from an inhibitor of tumor cell growth to a stimulator of growth and invasion in advanced stages of CRC [83]. An extensive meta-analysis concluded that high expression of TGF-β was a prognostic indicator in CRC patients undergoing surgery. The mortality rate of patients with a high expression of TGF-β was higher than that of patients with a low expression [83]. Thus, TGF-β could be a valuable prognostic biomarker in CRC. Inhibition of TGF-β signaling prevents metastasis or further development of certain advanced tumors such as CRC [84].

### 4.4. Interferon Gamma (IFN-γ)

IFN-γ plays a dual and opposing role in cancer development. On the one hand, IFN-γ signaling inhibits tumor growth, and on the other hand IFN-γ contributes to tumor growth through the promotion of tumorigenesis and angiogenesis [85]. Several publications have shown that CTLA-4 and PD-1 inhibitors, as well as other immune checkpoint inhibitors, result in increased IFN-γ production, which in turn leads to the killing of cancer cells [85]. Resistance to immunotherapy is attributed to defects in IFN-γ signaling [86].

### 4.5. Tumor Mutational Burden (TMB)

TMB is defined as the number of somatic mutations detected per megabase of tumor DNA. Its detection is of great relevance to increase the population benefiting from clinical immunotherapy [87]. A high TMB increases the probability of neoantigen generation; neoepitopes produced from mutated genes, when bound to major histocompatibility complex, are not recognized by T cells, leading to an effective antitumor immune response [88]. Higher TMB is associated with stronger immunogenicity, which could probably enhance the antitumor activity of immunotherapies. Of note is that high TMB overlaps with other biomarkers like MSI and TILs [89].

### 4.6. Tumor Infiltrating Lymphocytes (TILs)

TILs are lymphocytes located in the inflammatory infiltrates present in tumor islets and in the peritumoral stroma of solid tumors and are composed of cytotoxic T lymphocytes (TCD8), NK cells, and T helper lymphocytes (TCD4). In CRC, several studies support the prognostic value of the density of infiltration by TILs, depending on the specific subtype of lymphocytes that compose them [90]. Thus, the higher frequency of TCD8 and NK effector cells in tumor islets and peritumoral tissue seems to be associated with better long-term survival [89]. In addition to prognostic information, evaluation of TILs in locally advanced rectal cancer may help predict the degree of response to neoadjuvant chemoradiotherapy [91].

### 4.7. Tumor-Associated Macrophages (TAMs)

TAMs have an impact on the prognosis and efficacy of chemotherapy and immunotherapy [84]; they are mainly recruited from the periphery by chemokines released from tumor tissues. Such factors bind to corresponding receptors for monocyte/macrophage recruitment [81]. TAMs play an important role in promoting tumor angiogenesis and express a variety of membrane-bound molecules [81]. TAMs can be divided into M1-like and M2-like, which are shown to have antitumor and protumor activity in TME, respectively [92].

## 5. Markers of Response to Radiotherapy

There is strong evidence to recommend neoadjuvant radiotherapy for patients with clinical stage II–III rectal cancer [11,93]. Thus, a predictive model of response to radiotherapy, used in the pre-treatment stage, is critical to personalizing rectal cancer treatment and would facilitate organ preservation, perhaps even in patients for whom initial radiotherapy would not be routinely considered [94]. Some reports have described that the MRE11/RAD50/NBS1 (MRN) protein complex, involved in detecting and repairing DNA, may play an important role in various tumors, including CRC. The expression of MRN seems to be significantly associated with overall survival and disease-free survival in rectal cancer patients, including those treated with neoadjuvant radiotherapy. Furthermore, authors have proposed utilizing the MRN pathway to improve radiosensitivity in CRC patients [95,96]. Nevertheless, despite new biological insights and therapeutic advances, little is known about potential biomarkers capable of predicting pathologic tumor response prior to treatment and subsequently affecting patient prognosis [97]. Some of the markers of response to radiotherapy are shown in Table 2.

Over the past few years, preclinical and clinical studies have supported the rationale for integrating radiotherapy–immunotherapy. Radiotherapy can enhance the effects of immunotherapy by improving tumor antigen release, antigen presentation, and T-cell infiltration [103]. Radiotherapy and immunotherapy are more effective treatments in patients with low-volume disease. Patients with oligometastatic disease represent a subset of patients with metastatic cancer in whom the disease burden is limited. These patients may be in an ideal position to receive the greatest benefit from radioimmunotherapy. Therefore, future studies using radioimmunotherapy should focus on patients with oligometastatic disease [103]. Chemoradioimmunotherapy has shown to be effective and safe in patients with advanced dMMR/MSI-H CRC [104].

The addition of poly-ADP ribose polymerase (PARP) inhibitors to these genotoxic treatments is also being actively investigated. This presents an opportunity to combine immunotherapy and radiotherapy with PARP inhibition to improve patient responses and outcomes. Radiation therapy has radiosensitizing potential when combined with PI3K and PARP inhibitors. By combining these inhibitors with radiation and immunotherapy, doses of the agents can be reduced, which could reduce chemoresistance and dose-limiting toxicities [105].

## 6. Determination of Tumor Markers

For the determination of tumor markers, it is necessary to consider several aspects. On the one hand, there is the sample on which the determination is going to be made, such as serum, plasma, stool, exosomes, circulating tumor cells (CTCs), or circulating free DNA (cfDNA). On the other hand, there is the type of marker to be detected, such as genetic mutations, methylations, or miRNAs, among others. Finally, there is the technique to be used for the detection of these markers. Some of the most common samples, types of markers, and techniques are described below.

### 6.1. Sample Types

#### 6.1.1. Liquid Biopsy (LB)

LB is a procedure used for the detection of circulating tumor features in biological fluids, including CTCs, cfDNA derived from tumor (ctDNA), circulating miRNAs, exosomes, proteins, circulating messenger RNA (mRNA), long non-coding RNAs (lncRNAs), and tumor-educated platelets [106,107,108,109,110,111,112]. Analysis of these components can provide a real-time picture of tumor-associated changes [107]. In addition, these analytes reflect the cellular and molecular heterogeneity of tumors, unlike tissue biopsy, which only evaluates part of the cancer [110]. LB can provide additional information useful for diagnosis (screening and early detection), as well as prognostic assessment data (by detecting minimal residual disease (MRD)), ancillary staging, predictive assessment, risk of metastatic relapse, molecular profiling, estimation of risk of relapse, screening and monitoring of response to anticancer treatments, and identification of resistance mechanisms [109,110,112,113,114].

LB provides several benefits. One of the most promising is its ability to overcome the problem of tumor heterogeneity [109]. The temporal heterogeneity detected by LB is a good predictor of secondary resistance [114]. LB may be particularly useful in the treatment of CRC patients to identify recurrence (e.g., RAS mutation testing to detect the emergence of treatment resistance associated with anti-EGFR therapy), and for early detection of cancer in defined subpopulations, such as those at high risk for CRC [34]. Several studies have shown that circulating miRNAs can be used as LB biomarkers with relatively high sensitivity and specificity for early detection of various gastrointestinal cancers, such as CRC [115]. LB is more representative than tissue biopsy of the entire lesion and allows tumor evolution to be tracked in real time [114], with attention to metastatic CRC; moreover, the spatial omni-comprehensiveness of LB may outperform tissue biopsy as a more accurate tool for high-burden tumors [113]. In metastatic disease, LB can provide prognostic information by measuring tumor volume and can help predict tumor sensitivity to targeted therapies by detecting target mutation or monitoring tumor volume change [114]. In the first-line treatment setting, LB could accelerate molecular profile assessment for the administration of targeted anticancer agents (i.e., anti-EGFR drugs) [113]. LB also provides molecular information at the time of second-line chemotherapy selection [114]. The main strengths exclusive to LB are the detection of spatial and temporal tumor heterogeneities, the detection of MRD, and the monitoring of molecular volume change [114]. Table 3 shows the main advantages and disadvantages of LB.

Postoperative ctDNA detection provides evidence of MRD and identifies patients at very high risk of relapse among those with CRC who undergo curative surgery [114]. Zhang et al. observed that both colon and rectal cancer could be detected by ctDNA, and the latter had lower median plasma cfDNA plasma levels than patients with colon cancer (14.2 ng/mL vs. 8.94 ng/mL) [117]. ctDNA and prognostic levels could become additional decision criteria for reduction/intensification of initial baseline chemotherapy and modulation of postoperative treatment in the oligometastatic setting, even if the ideal cut-off point in this regard is missing [113].

Currently, LB results must be combined and evaluated with tissue pathological findings before final validation of the proposed approach [118]. Standardized pre-analytical methodologies need to be established in large prospective clinical studies, including blood collection, processing, and storage, as well as DNA extraction, quantification, and validation [110,119]. International consortia such as the European LB Society (ELBS, www.elbs.eu, accessed on 22 February 2023) can play an important role in this effort [120]. Vymetalkova et al., through a systematic review, concluded that LB should be considered key to the introduction of personalized medicine and subsequent patient benefits [119]. In the last few decades, LB has been postulated as a promising minimally invasive tool for cancer management and has led to the development of techniques to determine circulating biomarkers.

The accuracy of LB is facilitated by molecular techniques such as next-generation sequencing (NGS) or polymerase chain reaction (PCR)-based methods [116]. The combination of multiple biomarker tests should be the future guideline in CRC detection to increase their sensitivity and specificity [110]. Analysis of cfDNA, ctDNA, miRNA, or CTCs, among others, can be used for early cancer detection, ancillary staging, prognostic and drug resistance assessment, and MRD [110].

#### 6.1.2. cfDNA and ctDNA

Detection of cfDNA is probably the most promising procedure for the identification of MRD, evaluation of treatment response and prognosis, and identification of mechanisms of treatment resistance [119]. The detection rate of ctDNA depends on tumor type and volume [121]. ctDNA can be detected in 50% of patients with non-metastatic CRC and in almost 90% of patients with metastatic disease [122]. As the tumor grows, ctDNA release increases, which is associated with poor prognosis [107]. Furthermore, the presence of ctDNA, especially after surgery, has been associated with an increased risk of relapse [123,124]. Due to its small size, low content, and easy combination with plasma proteins, highly sensitive and reproducible techniques are needed for ctDNA extraction and genotyping, such as digital PCR (dPCR), amplification of refractory mutation system (ARMS), and NGS. However, the low proportion of fragments with mutations limits detection capability, resulting in false negatives. False positive results can occur for non-malignant mutations in hematopoietic cells [110,125].

### 6.2. Biomarker Types

#### 6.2.1. miRNA

Small non-coding RNAs, such as miRNAs, can be detected in serum and plasma and can be used as biomarkers to predict CRC patient survival, tumor stage, the presence of lymph node metastases, and response to therapy [126]. The method of choice for miRNA quantification is quantitative reverse transcriptase PCR (RT-qPCR) due to its high sensitivity and specificity. However, there is still a need to optimize and standardize methodologies for the assessment of circulating miRNA. Several clinical trials are evaluating the use of circulating miRNA as biomarkers in CRC [127].

#### 6.2.2. CTC

CTCs play an important role in oncogenesis and are involved in cancer cell proliferation, migration, and apoptosis. According to a meta-analysis, it is suggested that the detection of CTCs in peripheral blood by RT-PCR is a poor prognostic factor for patients with non-metastatic CRC and could be an early indicator of metastatic disease [128]. In recent decades, a number of techniques have been developed to isolate individual CTCs in blood. These technologies are based on biological or physical differences between CTCs and non-tumor blood cells. However, isolating CTCs remains challenging due to their infrequency and heterogeneity [129]. The US Food and Drug Administration (FDA) has only authorized the use of a CTC measurement platform (CellSearch^®^) that combines immunomagnetic enrichment with immunocytochemical and multiparametric flow cytometry analysis as a prognostic predictor of disease-free survival in metastatic CRC. CTCs can be a source of valuable tumor markers. There are tests to detect the expression of certain proteins (e.g., AR-V7) or the methylation profiles of tumor suppressor genes in isolated CTCs to select treatment and determine metastatic potential [129].

### 6.3. Biomarker Measurement Techniques

#### 6.3.1. NGS

NGS is used to identify all types of mutations and chromosomal abnormalities and thus provide molecular justification for appropriate targeted therapy. Its main advantages are that it does not require prior knowledge of the nature of possible genetic changes in the tumor and its high performance. Its main limitations are its relatively low sensitivity and its high economic cost. According to the Spanish Society of Medical Oncology (SEOM) and the Spanish Society of Pathological Anatomy (SEAP), the use of NGS in the study of CRC will allow for the diagnosis of Lynch syndrome through the mutational study of the *MMR* and *EPCAM* genes. In addition, NGS will guide treatment by detecting mutations in the main genes involved in CRC, such as *RAS*, *BRAF*, or *HER2*. This technique will also allow for the identification of tumor hypermutation status and molecular subtypes (CMS1, CMS2, CMS3, and CMS4), offering the possibility of designing personalized treatments. Thanks to the application of NGS to the study of ctDNA, it will be possible to monitor the response to treatment, anticipate the appearance of metastases, and detect resistance to ongoing treatment [1].

#### 6.3.2. PCR

Thanks to its high sensitivity, rapidity, and cost-effectiveness, PCR has been shown to be very useful in the identification of specific biomarkers [130]. Its main disadvantage is the limited ability to investigate a larger number (and different types) of genomic alterations [106]. However, dPCR allows for absolute quantification of target sequences and offers even greater sensitivity, being able to detect extremely rare sequences, but the cost of this technique is still higher than that of traditional quantitative PCR [131]. The ARMS system is a modification of the PCR technique that allows for the detection of any mutation involving single base changes or small deletions. It is commonly used in clinical laboratories, with acceptable accuracy and low cost [106]. The NGS, dPCR, and ARMS techniques have demonstrated high accuracy in detecting *KRAS* mutation in LB samples and could be used to guide anti-EGFR therapy in CRC patients without available tumor tissue samples [106].

## 7. Molecular Tumor Board (MTB)

Due to the disparity between clinical knowledge and genetic potential in cancer care, the implementation of multidisciplinary MTBs has been suggested to help clinicians in the interpretation of the massive amounts of information provided by genomic technologies and the determination of biomarkers [132].

Most MTBs consist of a multidisciplinary team of medical oncologists, surgeons, genetic counselors, pharmacists, pathologists, radiologists, bioinformaticians, and molecular biologists. They should also include bioethicists, at least when experimental drugs are used. Other professionals, such as scientists/physicians with a strong molecular background, can enhance the overall expertise related to MTBs and multidisciplinary discussions in the context of precision oncology. Since drugs can be proposed in clinical trials, a research/clinical trial coordinator can complete the integration of MTBs [4,7].

MTBs seek to translate increasingly complex genetic information into patient-centered clinical decisions, bringing precision oncology into daily practice. The establishment and organization of an MTB is critical, but there are currently no standards, guidelines, or quality requirements [132]. To successfully implement an MTB and to optimize its performance, as well as improve the interpretation and application of genomics-guided cancer care, it is necessary to achieve global harmonization in cancer sequencing practices and procedures, establish minimum membership and operational requirements, and put in place an appropriate unsolicited findings policy [132]. Key areas that should always be present in MTB reports are patient identification, reporting style and content (concise reporting and clear presentation of results), and interpretation of results [4].

An effective MTB must address critical issues and thus improve cancer management based on targeted therapies. Other important issues to be addressed in MTBs include (1) the acquisition of sufficient tissue at the initial diagnostic biopsy; (2) the reduction of preanalytical error levels and the adoption of highly standardized methodologies for tissue sampling/analysis; (3) technology to enable videoconferencing, reduce travel times for meetings, and allow consultation of extra-institutional experts; and (4) hospital management involvement in the implementation of prospective strategies to reduce the overall costs of precision medicine [4].

As larger cohorts of data become available and shared, it will be imperative to standardize the components of an MTB, such as the definition of actionability, off-label drug use, and types of sequencing [7]. The prospective randomized phase II SHIVA trial compared tumor molecular profiling-based therapy with conventional therapy in refractory cancer patients but unfortunately ended with negative results [133]. Another study showed that molecular-guided extended personalized patient care is effective in a small but clinically significant proportion of patients in challenging clinical situations [134]. Another work showed that patients receiving MTB-recommended regimens compared to the treating physician’s choice have significantly longer overall survival and progression-free survival and adapt better to therapies [3].

A well-designed MTB will evolve along with technology to ensure that patients receive the best possible treatment without unnecessary cost or risk and that physicians obtain ongoing educational information to help guide their decisions [3].

## 8. Conclusions

The determination of biomarkers in CRC is essential for choosing the best therapy for each patient, for establishing the prognosis of the disease, and for predicting the response to treatments. However, in order to carry out all of this, it is necessary to understand not only the involvement of this biomarker in tumor development but also which techniques currently exist to determine the involvement. These techniques provide a large amount of information that, without adequate knowledge of them, makes interpretation of them very difficult. To facilitate this task, MTBs have emerged, whose experience allows for accurate and up-to-date confirmation of diagnoses and for the identification of mutations and associated drugs together with the ability to recruit patients for open clinical trials.

## Figures and Tables

**Table 1 cancers-15-04373-t001:** Recommendations for the determination of CRC biomarkers.

Biomarker	Determination Recommendations
RAS	RAS mutation testing is recommended in patients with mCRC at the time of diagnosis, which allows for the determination of a prognosis and treatment, especially the use of anti-EGFR (and in the future RAS targeted therapy) [1,20]. In RAS WT disease that has received anti-EGFR antibodies and shows progression and for which maintenance or rechallenge to anti-EGFR is considered, re-analysis of RAS mutations is recommended [33]. Mutational analysis should include HRAS, KRAS, and NRAS and exons 2 (codons 12 and 13), 3 (codons 59 and 61), and 4 (codons 117 and 146) in tissue or in liquid biopsy [1,34]. It is not routinely recommended in non-metastatic patients [12].
BRAF	BRAF mutation testing is recommended in patients with mCRC in tissue at diagnosis for prognostic stratification [34,36] and to define treatment [10,11,36]. In tumors with dMMR, BRAF determination helps to evaluate the presence of Lynch syndrome. BRAF V600E mutation is consistent with sporadic cancer [34].
MSI	MSI analysis at diagnosis is recommended for all patients with advanced CRC, as it has an important prognostic and predictive role. It allows for the identification of a group of patients who would benefit from anti PD-1 immunotherapy [1]. In patients with stage II CRC, MSI predicts a low benefit of adjuvant fluoropyrimidine monotherapy. The ASCO guideline recommends that in the absence of risk factors, no adjuvant should be given to patients with MSI, and in those with risk factors, they should be given fluoropyrimidine + oxaliplatin [40]. In patients with stage III CRC, the prognostic and predictive value is less clear [1]. In these cases detection is limited to identifying Lynch syndrome [12].
PD-L1	There are no data to date to recommend the routine determination of PD-L1 in patients with mCRC. For now, its use should be limited to the research setting.
PI3K	PIK3CA determination is only recommended for the inclusion of patients in a clinical trial. It could play a potential role in chemoprevention.
FGFR	There are insufficient data to recommend its determination in mCRC in practice, except in the context of clinical research.
HER2	Its determination is recommended in patients with mCRC as a predictive factor for anti-EGFR therapy (first line), and in patients with left mCRC and RAS WT with progression to anti-EGFR therapy, with anti-HER2 therapeutic criteria.
NTRK	ESMO recommendations for using NGS recommend including NTRK testing [52]. According to local guidelines regarding an enriched population for determination, MSI-H and RAS WT is recommended with a progression to standard therapy.
RET	Studied in patients with advanced CRC, MSI-H, and RAS WT with resistance to immunotherapy.
ALK	Genomic profiling and NGS are recommended to study ALK. Another platform should be used depending on its availability and cost-effectiveness.
ROS1	Studied in patients with advanced CRC, MSI-H, and RAS WT with resistance to immunotherapy. Genomic profiling and NGS are recommended to study ROS1. Another platform should be used depending on its availability and cost-effectiveness.
NRG1	Approved for tumor-agnostic treatment (not in Latin America). Determination only under clinical trial.
MET	Study of amplification or mutation in cases of disease resistant to EGFR inhibitors, its prognostic role, and/or eventual study for participation in a clinical trial.
WEE1	Recent evidence from a phase II study with adavosertib [61]. Requires validation for determination as a predictive factor. Only in the context of clinical trials.

ASCO: American Society of Clinical Oncology; dMMR: DNA mismatch repair deficiency; ESMO: European Society for Medical Oncology; mCRC: metastatic colorectal cancer; MSI: microsatellite instability; MSI-H: microsatellite stability high; NGS: next-generation sequencing; WT: wild type.

**Table 2 cancers-15-04373-t002:** Markers of response to radiotherapy.

Marker	Predictive and Prognostic Value to Radiotherapy
Circulating tumor-specific DNA (ctDNA)	Neuropeptide Y gene hypermethylation (meth-ctDNA) could be a potential prognostic marker in the neoadjuvant setting and could be validated and identify patients at increased risk of distant metastasis [98]. ctDNA functions as a real-time indicator that can accurately reflect tumor burden [99].
Peripheral blood leukocytosis and neutrophilia	Associated with unfavorable clinical outcome in renal cancer patients treated in the phase III CAO/ARO/AIO-04 trial [100].
Circulating lymphocyte counts	Decreases during neoadjuvant therapy for locally advanced rectal cancer and is associated with improved tumor regression. It may be involved in the immune response elicited by radiotherapy and chemotherapy [101].
Carcinoembryonic antigen (CEA)	It is an independent predictor of tumor response to neoadjuvant treatment in patients with rectal cancer [102].

**Table 3 cancers-15-04373-t003:** Advantages and disadvantages of liquid biopsy.

Advantages	Disadvantages
It is a minimally invasive procedure with fast turnaround time and the ability to provide a more complete molecular picture of the disease, with low cost and minimal pain and risk [106,114]. It is considered a safe procedure [116]. Liquid biopsy allows for the feasibility of frequent repeat testing and extensive molecular characterization by depicting both spatial heterogeneity (intra-tumor and between different tumor sites) and temporal heterogeneity (mainly caused by cancer treatments over time) [113,116]. Serial liquid biopsy provides qualitative and quantitative information useful for assessing the risk of relapse by detecting minimal residual disease, for treatment selection, and for prognosis by measuring tumor molecular volume [114]. Since tumor re-biopsy is an invasive procedure (which may not be feasible), the only opportunity to investigate and monitor these alterations during treatment is liquid biopsy [109].	Contamination is a major problem in liquid biopsy [114]. False positive results can occur during the liquid biopsy detection phase due to the accumulation of benign circulating epithelial cells or blood cells [114].

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
