# Peer review of "Ibero-American Consensus Review and Incorporation of New Biomarkers for Clinical Practice in Colorectal Cancer"

_cancers, 2023, doi:10.3390/cancers15174373_

Round 1

Reviewer 1 Report

Thank you for the invitation to review the manuscript “Ibero-American expert consensus on biomarkers in colorectal cancer: Recommendations for Clinical Practice”.

In my opinion, this work has major limitations that should be seriously addressed before publication, as follows:

1.       The term “Ibero-American” is highlighted in the title, but the manuscript does not contain any Ibero-American-specific results or recommendations

2.       The authors should clearly differentiate between clinical recommendations (based on different levels of scientific evidence, which should be clearly stated in the article) and preliminary research findings with potential relevance for the future.

3.       I would strongly recommend that the authors focus on prognostic biomarkers; the current version of the article is a mixed description of risk, predictive and prognostic biomarkers.

4.       The manuscript would greatly benefit from adding some specific information for Spain and Latin America, even if sparse (e.g. mutation frequencies in LAT patients, inclusion of biomarker profiling and/or targeted treatment in national oncological practice, etc.)

5.       The simple summary needs to be revised. For example, imaging and FIT are not “genetic techniques”, and the article does not describe how CRC biomarkers are determined in Spain or Latin America.

6.       Abstract: Please focus on prognostic biomarkers. I cannot find in the article “the different points of view expressed by experts from Spain and Latin America”. Liquid biopsy and NGS are not different, but complementary techniques.

7.       Introduction: In the first sentence, the authors seem to confuse CRC risk factor heterogeneity (probably accessory to this paper) and tumour heterogeneity. How was the literature reviewed (e.g. what terms were used for the systematic review, or is this article purely expert opinion – if the second is the case, how were the different opinions combined?). The “Status map of biomarker determination” is not shown in the manuscript.

8.       Section 2: please use a more concrete subtitle, e.g. CRC mutations and MSI and show some specific information for Spain and Latin America.

9.       Table 1: Please add a column indicating the level of scientific evidence for each recommendation (on testing + treatment). The table needs to be heavily revised for consistency. For example, BRAF mutations are tested in tissue but RAS mutations are not? Evidence for PD-L1 and PI3K?

10.   Section 3: What is the current reality in Spain and Latin America, what should be prioritised in the near future?

11.   Section 4: Same question as in section 3. Why is TGF-beta listed as a biomarker for immunotherapy?

12.   Section 5: Is the long introduction on CRC radiotherapy necessary? Is CEA as marker of radiotherapy response?

13.   Section 6 is quite superficial and the current level of evidence is low, I would recommend deleting it or improving it comprehensively.  

14.   Section 7: Please focus on CRC prognosis, the utility of miRNAs for early CRC detection is probably secondary for this article. Please restructure this section according to sample types (e.g. serum, plasma, exosomes, cfDNA,…), biomarker types (mutations, methylation, miRNAs, lncRNAs,…) and biomarker measurement techniques (e.g. NGS, PCR, immunochemistry). Again, please highlight current practice and priorities in Spain and Latin America.

15.   Section 8: Same comments as above. Some ongoing experiences in Spain and Latin America?

In summary, I consider that the article as it stands has very little clinical value and I cannot recommend its publication. However, I would be willing to review a much improved version.

No additional comments

Author Response

REVIEWER 1

Thank you for the invitation to review the manuscript “Ibero-American expert consensus on biomarkers in colorectal cancer: Recommendations for Clinical Practice”.

In my opinion, this work has major limitations that should be seriously addressed before publication, as follows:

  1. The term “Ibero-American” is highlighted in the title, but the manuscript does not contain any Ibero-American-specific results or recommendations.

As we stated at the end of the Introduction, the objective of our work was to briefly describe the importance of biomarker determination in CRC based on the opinion and clinical experience of experts from Spain and several Latin American countries. There are no specific results or recommendations for these countries, but rather the opinions of experts based on their personal clinical experience and the published scientific evidence.

  1. The authors should clearly differentiate between clinical recommendations (based on different levels of scientific evidence, which should be clearly stated in the article) and preliminary research findings with potential relevance for the future.

Our work is not a review article or a consensus with levels of evidence and grades of recommendation, but a consensus of experts describing the importance of the determination of biomarkers in CRC according to their clinical experience. The objectives and how they were achieved are described in the abstract and at the end of the Introduction. However, we have modified the Abstract and the Introduction section to clarify the purpose of the article.

  1. I would strongly recommend that the authors focus on prognostic biomarkers; the current version of the article is a mixed description of risk, predictive and prognostic biomarkers.

Thank you for your kind proposal. We think that biomarker information is obviously very relevant but in our opinion the information about how to predict response to treatment is currently very helpful for a better management of these patients.

  1. The manuscript would greatly benefit from adding some specific information for Spain and Latin America, even if sparse (e.g. mutation frequencies in LAT patients, inclusion of biomarker profiling and/or targeted treatment in national oncological practice, etc.)

Although we believe that this would be valuable and important information, given the large number of participating countries, the extent of the epidemiological data would make the article exceed the limits of the journal, and could even lose the focus of biomarker determination.

  1. The simple summary needs to be revised. For example, imaging and FIT are not “genetic techniques”, and the article does not describe how CRC biomarkers are determined in Spain or Latin America.

According to the journal's instructions, the Simple Summary should use language that can be understood by common people, which could lead to certain issues not being expressed properly. Even so, and to clarify the reviewer's suggestion, we have made a slight modification to this section.

Taking into account that the article reflects the opinion and clinical experience of the participating experts, it can be understood that this is how CRC biomarker determination is done in their corresponding centres.

  1. Abstract: Please focus on prognostic biomarkers. I cannot find in the article “the different points of view expressed by experts from Spain and Latin America”. Liquid biopsy and NGS are not different, but complementary techniques.

Please, refer to question 3 to address the comments related to prognostic biomarkers in the abstract.

There is no specific section on the experts' points of view. The entire article is a reflection of what was discussed among them, supporting their comments with bibliographic references. However, as we have stated in question 2, we have modified the Abstract and Introduction to clarify the purpose of the article.

NGS is currently mainly used in the study of tumor tissue. On the other hand, liquid biopsy can be applied for monitoring patients using the analysis of one genetic alteration or assessing multi-gene panel. Both techniques can be used in different scenarios. With the growing amount of new biomarkers we are moving forward the use of NGS instead one single-gene analysis, and it is something that it is happening in the analysis applied in liquid biopsy where we need to analyze more and more genetic alterations in any time moment and in a continuous reassessment.

  1. Introduction: In the first sentence, the authors seem to confuse CRC risk factor heterogeneity (probably accessory to this paper) and tumour heterogeneity. How was the literature reviewed (e.g. what terms were used for the systematic review, or is this article purely expert opinion – if the second is the case, how were the different opinions combined?). The “Status map of biomarker determination” is not shown in the manuscript.

We have revised the first sentence of the introduction to avoid confusion. It now reads: “Colorectal cancer (CRC) is a heterogeneous disease that results from the interaction of multiple genetic modifications and exogenous factors such as diet, lifestyle, and micro-biome”.

This article is not a systematic review (so there is no standardized literature search methodology), but a consensus in which a group of experts met and discussed the importance of biomarker determination in CRC according to their own clinical experience and based on published scientific evidence. During the meetings, a series of topics for discussion were raised and, based on the most relevant literature on these topics, they were debated until the experts agreed. However, we have revised the Introduction to clarify the type of article that we are showing.

Regarding the status map of biomarker determination, we have modified the Introduction to describe the situation that clinicians have to face in the determination of biomarkers in CRC.

  1. Section 2: please use a more concrete subtitle, e.g. CRC mutations and MSI and show some specific information for Spain and Latin America.

We prefer to keep the title, since in this section we talk about tumor markers in general, either genetic or molecular alterations, and even the location of the tumor itself. However, we have made some minor modifications to the text to make it more coherent.

At the moment we are unable to provide specific data on biomarkers in patients from Spain and Latin America. We are analyzing the real frequency of MSI in patients from our countries to confirm that this is similar to previously reported in the main articles from US. In other biomarkers, for example in EGFR mutations, patients from some Latin American countries showed much higher frequency than expected.

  1. Table 1: Please add a column indicating the level of scientific evidence for each recommendation (on testing + treatment). The table needs to be heavily revised for consistency. For example, BRAF mutations are tested in tissue but RAS mutations are not? Evidence for PD-L1 and PI3K?

As mentioned in point 2, the aim of the article is not to establish levels of evidence and degrees of recommendation. The tables should only be considered as a support to the content of the text, showing the information in the most practical and simple way possible, but understanding that their content was chosen according to the opinion of the participating experts.

  1. Section 3: What is the current reality in Spain and Latin America, what should be prioritised in the near future?

The situation is very unequal in the different regions of Spain as well as the countries of Latin America. Initiatives have been launched in different hospitals through research funds, especially in some Latin American countries. The Spanish Government approved in 2020 the call for the Precision Medicine Infrastructure associated with Science and Technology (IMPaCT program) of the Strategic Health Action 2017-2020. The initiative has been a first step for the implementation of Precision Medicine in the National Health System through a strategy based on science and innovation.

  1. Section 4: Same question as in section 3. Why is TGF-beta listed as a biomarker for immunotherapy?

TGF-beta has been incorporated because we believe that it could help to identify molecular environments where immunotherapy alone or in combinations with some targeted drugs could identify cases with little or no response.

  1. Section 5: Is the long introduction on CRC radiotherapy necessary? Is CEA as marker of radiotherapy response?

We have substantially reduced the introduction in this section.

CEA has been postulated in some articles in relation to the response to radiation but its value is not confirmed.

  1. Section 6 is quite superficial and the current level of evidence is low, I would recommend deleting it or improving it comprehensively.

We agree with the reviewer. We have removed this section completely.

  1. Section 7: Please focus on CRC prognosis, the utility of miRNAs for early CRC detection is probably secondary for this article. Please restructure this section according to sample types (e.g. serum, plasma, exosomes, cfDNA…), biomarker types (mutations, methylation, miRNAs, lncRNAs…) and biomarker measurement techniques (e.g. NGS, PCR, immunochemistry). Again, please highlight current practice and priorities in Spain and Latin America.

As we stated in section 3, we think that biomarker information is obviously very relevant but in our opinion the information about how to predict response to treatment is currently very helpful for a better management of these patients.

According to reviewer’s suggestion, the new section 6 (previously section 7) has been restructured.

We appreciate the comment on the role of the different RNA molecules and we understand that perhaps we have not conveyed the objective that we pursue with this part. Both in Spain and Latin America very active research is being carried out on the role played by these molecules, especially RNA non-coding in the prediction of response to different treatments such as in the prognosis of colon and rectal cancer. We think that it could be a differential aspect in which data from Latin American research groups were exposed.

  1. Section 8: Same comments as above. Some ongoing experiences in Spain and Latin America?

Note that section 8 becomes section 7. We are very excited about the incorporation of these new biomarkers beyond the classical approach based on RAS, BRAF and MSI. These new options provided in this manuscript through these biomarkers can improve the management of our metastatic cancer patients.

Reviewer 2 Report

An excellent comprehensive review in the area of colorectal cancer should be published

Accept as is

No problems with the English

Author Response

REVIEWER 2

An excellent comprehensive review in the area of colorectal cancer should be published.

Accept as is.

Thank you very much for approving the manuscript.

Reviewer 3 Report

Good morning! I have no further questions for the author.

Author Response

REVIEWER 3

Good morning! I have no further questions for the author.

Thank you very much for approving the manuscript.

Reviewer 4 Report

The article is well-written and is a good review of the literature on colorectal cancer biomarkers. However, the authors should be more clearer about the biomarkers they are looking at i.e. predictive or prognostic biomarkers. Screening biomarkers such as FIT are not in the scope of this review.
The paragraph starting from line 119 – “Despite the fact that many of the biomarkers described are not routinely determined, it is important to mention them, even if only briefly, as they add value to the classical approach of biomarkers so far considered, which were reduced classically only to RAS, 121 BRAF, and MSI” should be rephrased to emphasise the prognostic/predictive nature of the biomarkers being reviewed.

The authors write starting in line 286 that “little is known about potential biomarkers capable of predicting pathologic tumor response prior to treatment and subsequently affecting patient prognosis [52]”. Strictly speaking this is not true as a number of important biomarkers were not covered with respect to rectal cancer radiotherapy. As an example see review article - https://www.ncbi.nlm.nih.gov/pmc/articles/PMC6413120/ for coverage of DNA damage response pathway proteins. Another paper https://www.ncbi.nlm.nih.gov/pmc/articles/PMC6122630/ describes a 3-protein panel of MRN complex as a potential radiosensitivity marker.

Minor comments relate to syntax:

In line 45 the sentence “article also describes the importance to look for additional prognostic biomarkers” should be reworded to “article also describes the importance of looking for additional prognostic biomarkers”

In line 221 the sentence “The mortality rate of patients with a high expression of TGF-β was higher that of patients with a low expression” should be reworded to “The mortality rate of patients with a high expression of TGF-β was higher than that of patients with a low expression”

As above

Author Response

The article is well written and is a good review of the literature on colorectal cancer biomarkers. However, the authors should be clearer about the biomarkers they are looking at i.e. predictive or prognostic biomarkers. Screening biomarkers such as FIT are not in the scope of this review.

  1. The paragraph starting from line 119 – “Despite the fact that many of the biomarkers described are not routinely determined, it is important to mention them, even if only briefly, as they add value to the classical approach of biomarkers so far considered, which were reduced classically only to RAS, BRAF, and MSI” should be rephrased to emphasize the prognostic/predictive nature of the biomarkers being reviewed.

Thank you very for your feedback. According to you suggestion, we have removed FIT section and other content related to FIT after Table 3. In addition, the paragraph mentioned at the end of section 2 has been modified to include the proposal suggested.

  1. The authors starting in line 286 that “little is known about potential biomarkers capable of predicting pathologic tumor response prior to treatment and subsequently affecting patient prognosis”. Strictly speaking this is not true as a number of important biomarkers were not covered with respect to rectal cancer radiotherapy. As an example see review article – https://www.ncbi.nlm.nih.gov/pmc/articles/PMC6413120/ for coverage of DNA damage response pathway proteins. Another paper https://www.ncbi.nlm.nih.gov/pmc/articles/PMC6122630/ describes a 3-protein panel of MRN complex as a potential radiosensitivity marker.

We have taken note of the suggestion and have added a paragraph with the information provided in section 5.

  1. Minor comments relate to syntax:

In line 45 the sentence “article also describes the importance to look for additional prognostic biomarkers” show be reworded to “article also describes the importance of looking for additional prognostic biomarkers”.

In line 221 the sentence “The mortality rate of patients with a high expression of TGF-β was higher that of patients with a low expression” should be reworded to “The mortality rate of patients with a high expression of TGF-b was higher than that of patients with a low expression”.

Thank you very much for the feedback. We have corrected both sentences with your suggestions.

Reviewer 5 Report

This is a V1 review and the authors have very carefully revised the manuscript. The sections with concern have been removed and new sections have been properly edited. It is reading well and overall, very informative. It gives in nut shell a comprehensive overview of the diagnostic and prognostic markers for CRC and how to interpret them to translate into identification of the staging of the cancer. Thus, this reviewer is excited about this manuscript.

English quality is moderately good.

Author Response

This is a V1 review and the authors have very carefully revised the manuscript. The sections with concern have been removed and new sections have been properly edited. It is reading well and overall, very informative. It gives in nut shell a comprehensive overview of the diagnostic and prognostic markers for CRC and how to interpret them to translate into identification of the staging of the cancer. Thus, this reviewer is excited about this manuscript.

We greatly appreciate your comments and support for the publication of the article.
